physical chemistry

molecular assembly, competitive adsorption, flotation separation

**Author for correspondence:**
Wei Xiao
e-mail: xiaowei2015@yahoo.com

# The effect of molecular assembly between collectors and inhibitors on the flotation of pyrite and talc

## Tao Long, Wei Xiao and Wei Yang

School of Resources Engineering, Xi'an University of Architecture and Technology, Xi'an 710055, People's Republic of China

  WX, 0000-0001-5245-4926

In the flotation process, the traditional dosing sequence is to add an inhibitor first, followed by a collector. However, in the sorting process of copper sulfide ore, this method of dosing does not effectively separate sulfide minerals and layered magnesium silicate minerals. In this study, the effect of adding a guar gum (as an inhibitor) and potassium amyl xanthate (as a collector, shortened as PAX) sequence to the flotation separation of pyrite and talc was investigated by micro-flotation tests, adsorption amount measurements, contact angle measurement and FT-IR analysis. The results show that the collector only adsorbs on the pyrite surface, while the inhibitor has a strong adsorption capacity on the pyrite and talc surface. Through the change of the order of the flotation reagent addition, PAX preferentially adsorbs on the pyrite surface, thereby preventing guar gum from adsorbing on the pyrite surface and achieving the selective inhibition of talc by guar gum. This study will help in understanding the molecular assembly between collectors and inhibitors to further treat complex copper sulfide nickel ore.

## 1. Introduction

The pre-enrichment of metal extraction in sulfide minerals is usually carried out by flotation [1]. The collector is used to increase the hydrophobicity of sulfide minerals, while the inhibitor improves the hydrophilicity of gangue minerals, and then the flotation method is used to separate sulfide minerals and gangue minerals based on the surface wettability between them [2–4]. Talc is a common refractory gangue mineral in sulfide ores. Due to its strong natural hydrophobicity, polysaccharides and cellulosic polymers are usually used as inhibitors to inhibit talc floating [5–7]. However, the inhibitors of organic polymers have a

problem of poor selectivity and have a strong inhibitory effect on the target mineral, making it difficult to efficiently separate the mineralized mineral from the talc [8,9].

Guar gum is a commonly used polysaccharide flotation inhibitor [10], which is used in the classification of silicate minerals such as sulfide minerals and talc, but the selectivity of this inhibitor is poor [11,12]. In recent years, researchers have carried out modification studies on guar gum [12–15] to enhance the selectivity of gangue mineral inhibition, or to replace guar gum with new agents [16,17], but the problem of the selective adsorption of reagent molecules has not been solved. Studies have reported that the interfacial assembly of reagents can improve the selectivity of flotation agents and enhance the separation efficiency of useful minerals and gangue minerals [18–20]. Bicak et al. [8] compared the inhibitory effects of guar gum and carboxy methyl cellulose (CMC) on pyrite in sulfide flotation and found that guar gum had a strong inhibitory effect on pyrite at low dosage. Therefore, under conventional flotation conditions, guar gum is difficult to use as an inhibitor of gangue minerals with better natural flotation, such as sulfide ore and talc. Wang et al. [21] used a dodecylamine/sodium oleate surfactant to flocculate and separate muscovite and quartz, and studied the self-assembly mechanism of the flotation agent and its inhibition mechanism on quartz. Jiao et al. [22] used a multi-confectionery gel to separate scheelite and calcite by flotation separation. The carboxyl group on the molecular chain of the pectin chemically chelated with the calcium ion on the calcite surface and adsorbed on the calcite surface. The large amount of pectin adsorption hinders the further adsorption of the collector sodium oleate on the calcite surface, which greatly reduces the floatability of the calcite, and finally achieves the flotation separation of the two minerals. Chen et al. [23] studied the molecular assembly characteristics of various inhibitors and sodium oleate on a mineral surface and realized the efficient flotation separation of fluorite and calcite with polyaspartic acid. Dong et al. [24] studied the molecular assembly of the inhibitor xanthan gum and the collector sodium oleate in the flotation separation of scheelite and calcite. The inhibitor xanthan gum adsorbed more on the surface of calcite and hindered the capture. The further adsorption of sodium oleate reduced the flotation of calcite and achieved the flotation separation of the two minerals. Tian et al. [25] studied the mechanism of action of the inhibitor ethylenediaminetetraacetic acid and the collector sodium lauryl sulfate in the flotation separation of lapis lazuli with fluorite and calcite. Ethylenediaminetetraacetic acid can be chelated. Adsorption on the surface of fluorite has a strong inhibitory effect on the dissolution of calcium ions on the surface of fluorite, thereby reducing the effect of dissolved calcium ions on the flotation of lapis lazuli.

In this paper, according to the difference of the adsorption characteristics of the collector and the inhibitor on the surface of the target mineral and the gangue mineral, the potassium amyl xanthate (PAX) preferentially adsorbs on the surface of the pyrite by the change of the order of addition of the flotation agent, thereby preventing the adsorption of the guar gum. On the surface of pyrite, a selective inhibition of guar gum against talc is achieved. This study will help in understanding the molecular assembly between the collector and the corrosion inhibitor to further process complex talc-type copper-sulfide–nickel ore.

# 2. Material and methods

## 2.1. Materials and reagents

The pyrite was obtained from Yunfu, Guangdong Province, China, and the talc was obtained from Haicheng, Liaoning Province China. According to X-ray diffraction (XRD, in figure 1) and elemental analyses (in table 1), the purities of pyrite and talc were 97.6% and 95%, respectively. The samples were dry ground and screened to different sizes used for different tests. The guar gum and PAX were purchased from Shanghai CIVI Chemical Technology Co., Ltd. In this study, all reagents were of analytical grade.

## 2.2. Methods

### 2.2.1. Micro-flotation tests

The flotation test uses a 40 ml XFG trough-type flotation machine. The ore sample weighing 2 g was placed in the flotation cell for each test. Then, a certain concentration of flotation reagent was added and stirred for 5 min. The pH value of the slurry was measured by a precision pH meter and floated.

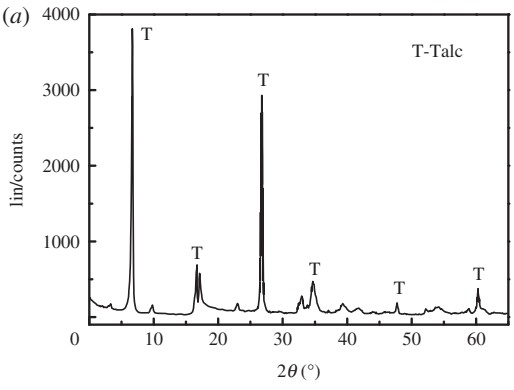 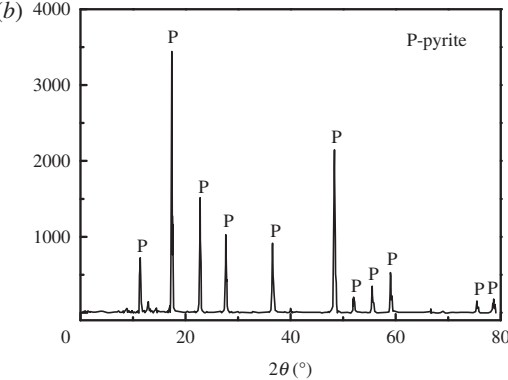

**Figure 1.** XRD pattern of talc (*a*) and pyrite (*b*).

**Table 1.** Multi-element analysis of pure talc and pyrite (%).

| sample | MgO | SiO$_2$ | Fe | S |
|---|---|---|---|---|
| talc | 31.66 | 63.70 | 1.01 | — |
| pyrite | — | — | 43.96 | 49.98 |

**Table 2.** The specific surface area of talc and pyrite.

| sample | talc | pyrite |
|---|---|---|
| specific surface area (m$^2$ g$^{-1}$) | 7.88 | 0.027 |

The flotation process was carried out by manual scraping. The flotation time was 5 min. The obtained foam product and the product in the tank were dried, weighed and the yield was calculated. The single mineral test takes the flotation recovery rate equal to the yield; in the artificial mixed ore flotation test, the flotation recovery rate is calculated after the chemical analysis of each product.

### 2.2.2. Adsorption amount measurements

According to the difference in the concentration of the flotation agent in the solution before and after the adsorption of the mineral, the adsorption amount of the potassium pentoxide and the guar gum on the mineral surface was determined by the residual concentration method.

The absorbance of the solution before and after the action of the flotation agent was measured by TU1810 ultraviolet–visible spectrophotometer. The characteristic absorption peak of PAX was at 301 nm, and the colour reaction of the guar gum solution was carried out by the phenol–sulfuric acid method. The characteristic absorption peak of guar gum is at a wavelength of 487.5 nm. First, the absorbance of the solution at different reagent concentrations was measured, and the working curve of the drug concentration and absorbance was plotted. Then, the absorbance of the flotation agent in the solution before and after adsorption on the mineral surface was measured, the absorbance was converted into the residual concentration of the agent in the solution to be tested by the working curve, and the adsorption amount of the potassium pentoxide and the guar gum on the mineral surface was calculated. Table 2 shows the specific surface area of each mineral sample.

### 2.2.3. Contact angle measurements

The single mineral lump ore was cut into $1 \times 2 \times 1$ cm$^3$ squares using a cutter, first polished with cast iron, then coarsely ground with Al$_2$O$_3$ abrasive, and finely ground with Cr$_2$O$_5$ abrasive. Before the test, the surface was polished with metallographic sandpaper, and then ultrasonically cleaned for 5 min. The agent was added according to the same conditioning conditions as the flotation test, and the ore sample was placed in the solution to soak, and for the time corresponding to the flotation was stirred.

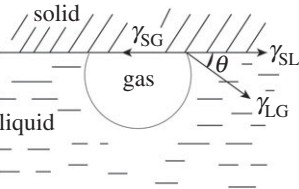

**Figure 2.** Schematic diagram of the contact angle measurement of mineral surface.

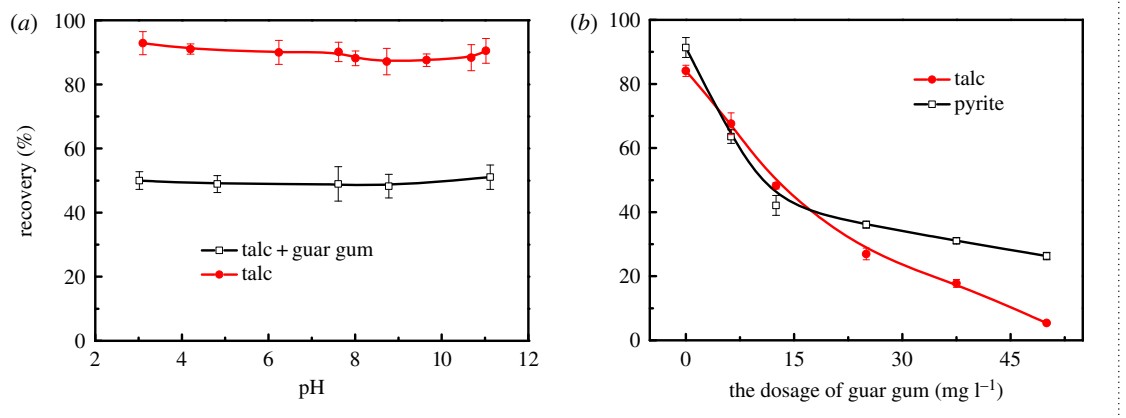

**Figure 3.** (a) The flotation recovery of talc as a function of pH in the presence of 12.5 mg l$^{-1}$ guar gum; (b) the flotation recovery of talc and pyrite as a function of guar gum dosage at pH 9.0. [MIBC] = 10 mg l$^{-1}$.

The contact angle $\theta$ was measured by the bubble method, as shown in figure 2. The measuring instrument used was a JJC-1 type wet contact angle measuring instrument.

### 2.2.4. FT-IR spectroscopy

We weighed 2 g of a single mineral sample with a particle size of less than 2 μm, added a certain concentration of flotation agent and stirred well and then let the mixture stand for a while. After the mineral was completely settled, the supernatant liquid was sucked out by a pipette. The mineral was thoroughly washed with distilled water, and dried naturally after solid–liquid separation. Infrared spectroscopy was carried out using a Nicolet FTIR-740 Fourier transform infrared spectrometer.

# 3. Results

## 3.1. Micro-flotation tests

Due to the natural hydrophobicity of talc, in the flotation separation of sulfide ore and talc, it is necessary to add a talc inhibitor and sulfide ore collector. Guar gum is a highly effective inhibitor of talc. Figure 3a shows the effect of guar gum on the floatability of talc at different pH values. As shown in the figure, after adding 12.5 mg l$^{-1}$ guar gum, the flotation recovery rate of talc decreased from 90 to 50%, and the shale floatability was significantly reduced. The inhibitory effect of guar gum on talc is not affected by pH.

Figure 3b shows the effect of guar gum dosage on talc and pyrite flotation at a pH of 9. With the increase in the dosage of guar gum, the recovery of talc and pyrite decreased rapidly. When the dosage of guar gum was 50 mg l$^{-1}$, the recovery rate of talc was only 5%. At this time, the flotation of talc was completely inhibited. The guar gum colloid has a branched structure. After adsorption on the surface of the talc, the guar gum molecule has a ring structure and a tail extending to the water phase. This part of the extension structure can strongly shield the talc from the bubble adhesion, thereby strengthening the inhibition of talc. It can be seen from the figure that talc and pyrite are strongly inhibited as the amount of guar gum is increased. However, when the dosage of guar gum was 50 mg l$^{-1}$, the recovery of pyrite was only below 30%. This indicates that guar gum cannot be used as an inhibitor of the flotation separation of talc and pyrite when there is no collector.

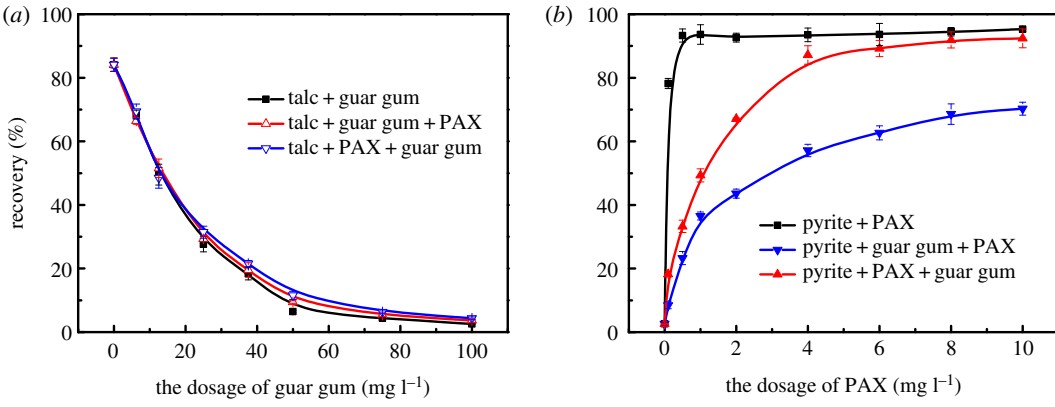

**Figure 4.** The flotation recovery of talc and pyrite as a function of guar gum (*a*) and PAX (*b*) dosage.

**Table 3.** The results of flotation tests of the artificial ore.

| reagent system[a] | produce | rate (%) | grade (%) | | recovery (%) | |
|---|---|---|---|---|---|---|
| | | | S | MgO | S | MgO |
| PAX | concentration | 85.42 | 16.22 | 13.84 | 88.23 | 81.20 |
| | tailing | 14.58 | 12.67 | 18.78 | 11.77 | 18.80 |
| | feed | 100 | 15.70 | 14.56 | 100.00 | 100.00 |
| guar gum + PAX | concentration | 49.72 | 26.17 | 2.70 | 82.89 | 9.22 |
| | tailing | 50.28 | 5.34 | 26.29 | 17.11 | 90.78 |
| | feed | 100 | 15.70 | 14.56 | 100.00 | 100.00 |
| PAX + guar gum | concentration | 48.96 | 27.67 | 1.23 | 86.28 | 4.14 |
| | tailing | 51.04 | 4.22 | 27.35 | 13.72 | 95.86 |
| | feed | 100 | 15.70 | 14.56 | 100.00 | 100.00 |

[a][PAX] $= 2 \times 10^{-4}$ mol l$^{-1}$; [guar gum] $= 50$ mg l$^{-1}$; at pH 9.0.

Since guar gum has a strong inhibitory effect on talc and pyrite, we investigated the effect of the order of addition of collectors and guar gum on flotation recovery. The results are shown in figure 4. Figure 4*a* shows the effect of the flotation agent on the floatability of talc at the mineral solid–liquid interface. When there is no inhibitor, the talc has good floatability, and the flotation recovery rate reaches 84%. In the system where the flotation agent and the inhibitor coexist, the floatability of the talc is strongly inhibited. When the amount of guar gum is 50 mg l$^{-1}$, talc is almost completely inhibited.

Figure 4*b* shows the effect of the flotation agent on the floatability of pyrite in the intermolecular assembly of the mineral solid–liquid interface. When there is no inhibitor, the floatability of pyrite is very good, and the flotation recovery rate is over 90%. In the system where the flotation agent and the inhibitor coexist, the floatability of pyrite is inhibited to some extent. The floatability of pyrite is restored after the intermolecular assembly of surface flotation agents. The intermolecular assembly of the flotation agent at the mineral solid–liquid interface increases the floatability of the sulfide ore, reduces the floatability of the talc and enhances the floatability difference of the sulfide ore and talc.

Through the flotation experiment of pyrite and talc artificial mixed ore, the flotation separation effect of pyrite and talc after the intermolecular assembly of the solid–liquid interface flotation agent was investigated. The artificial mixed ore flotation test was carried out in a 40 ml hanging trough flotation machine. The flotation time was 3 min, the amount of artificial mixed ore was 2 g, and the ratio of pyrite to talc was 1 : 1. PAX was used as the collector, and guar gum was used as the inhibitor.

Table 3 shows the enhancement of the separation of pyrite and talc from the intermolecular assembly of guar gum and PAX. According to the data in table 3, when the inhibitor is not used, the S grade and MgO contents in the concentrate are very high, both pyrite and talc enter the concentrate, and there is no flotation separation effect. After the inhibitor is added, the grade of S in the concentrate increased from

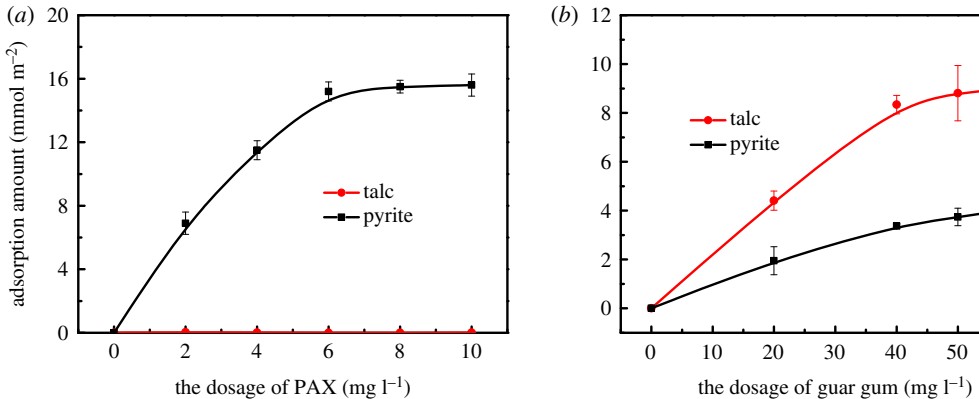

**Figure 5.** The adsorption amount of PAX (*a*) and guar gum (*b*) on talc and pyrite surfaces as a function of PAX and guar gum, respectively. PAX and guar gum exist separately.

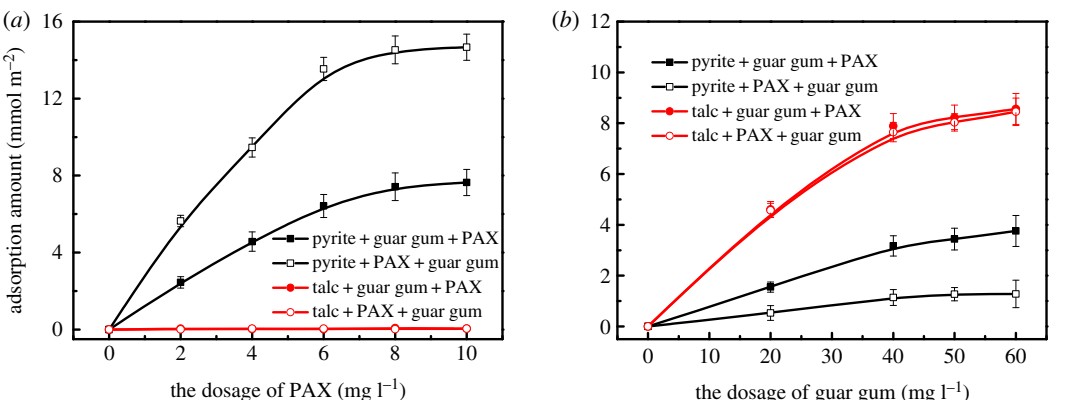

**Figure 6.** The adsorption amount of PAX (*a*) and guar gum (*b*) on talc and pyrite surfaces as a function of PAX and guar gum. PAX and guar gum exist simultaneously.

16.22 to 26.17%, and the MgO content decreased from 13.84 to 2.70%, but the recovery of concentrate S decreased from 88.23 to 82.89%. This indicated that pyrite was inhibited to some extent. After the flotation agent intermolecular assembly, the S grade in the concentrate was further increased from 26.17 to 27.67%, the MgO content was further reduced from 2.70 to 1.23% and the recovery rate of concentrate S was increased from 82.89 to 86.28%. The mineral–solid–liquid interface guar gum and PAX intermolecular assembly strengthens the flotation separation of sulfide ore and talc.

## 3.2. Adsorption amount measurements

The adsorption amount of the flotation reagent on the mineral surface as a function of dosage is shown in figure 5. Figure 5*a* shows the adsorption isotherms of the collector pentyl potassium xanthate (PAX) on the surface of pyrite and talc, respectively. It can be seen that with the increase in the amount of PAX, the adsorption amount of PAX on the surface of pyrite gradually increases and eventually stabilizes. While PAX is not adsorbed on the surface of talc, it is suggested that the adsorption of PAX on the surface of two minerals has a good selectivity.

Figure 5*b* shows the adsorption isotherms of the inhibitor guar gum on the surface of pyrite and talc, respectively. It can be seen that guar gum is adsorbed on the surface of pyrite and talc. Although the adsorption amount of guar gum on the surface of talc is larger than that on the surface of pyrite, the adsorption amount of guar gum on the surface of pyrite is also large. This shows that the adsorption of guar gum on the surface of these two minerals is poor.

When PAX and guar gum exist simultaneously, the adsorption amount of PAX and guar gum on talc and pyrite surfaces as a function of PAX and guar gum is shown in figure 6. Figure 6*a* shows the adsorption isotherms of PAX on the pyrite and talc surface before and after the intermolecular assembly of flotation agents. It can be seen that the adsorption capacity of PAX on the surface of

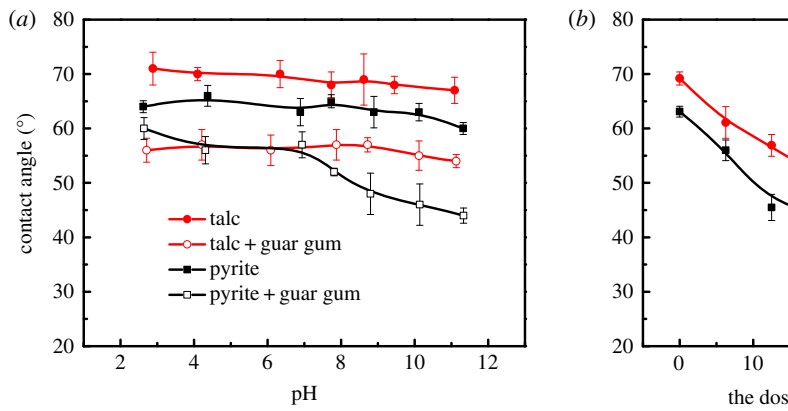

**Figure 7.** The contact angle of talc and pyrite surfaces as a function of pH (*a*) and guar gum dosage (*b*) in the absence of a collector.

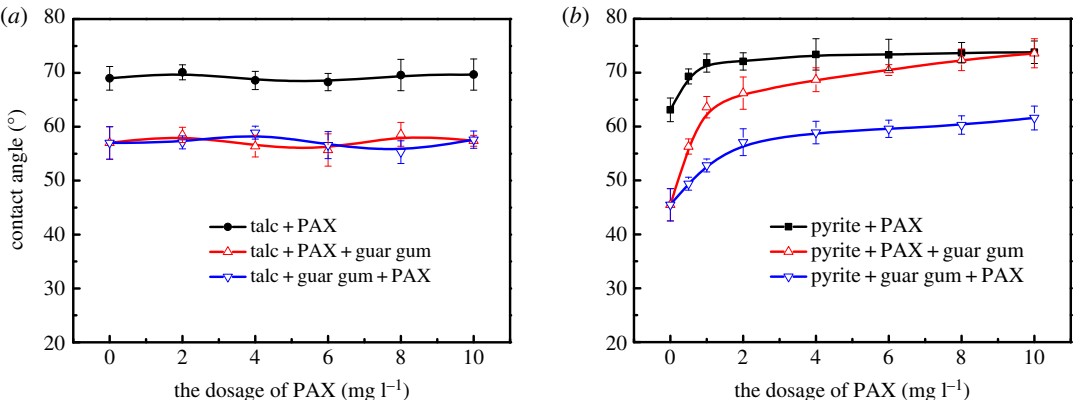

**Figure 8.** The contact angle of talc (*a*) and pyrite (*b*) surfaces as a function of PAX dosage. [guar gum] = 12.5 mg l$^{-1}$, at pH 9.0.

pyrite is significantly increased after the flotation agent is assembled. However, talc still does not adsorb the collector xanthate after the flotation agent is assembled between molecules.

Figure 6*b* shows the adsorption isotherms of guar gum on the pyrite and talc surface before and after the intermolecular assembly of flotation agents. It can be seen that the adsorption of guar gum on the surface of pyrite is remarkably reduced after the intermolecular assembly of flotation agent. The adsorption capacity of guar gum on the talc surface before and after the intermolecular assembly of the flotation agent did not change significantly. The adsorption of the talc on the talc surface was maintained. This indicated that the intermolecular assembly of the flotation agent enhances the selectivity of the inhibitor guar on the surface of the two minerals. The intermolecular assembly of the collector and the inhibitor on the mineral surface enhances the adsorption of the collector on the surface of the target mineral and enhances the selectivity of the inhibitor on the surface of the gangue mineral.

## 3.3. Contact angle measurements

Figure 7*a* shows the effect of guar gum on the surface wettability of talc and pyrite at different pH conditions. After the addition of guar gum, the contact angles of the talc and pyrite surface become smaller and the hydrophobicity deteriorates. Under alkaline conditions, pyrite is more effective than talc. Figure 7*b* shows the effect of guar gum dosage on the surface wettability of talc and pyrite at a pH of 9. With the increase in the amount of guar gum, the contact angle of talc and pyrite gradually became smaller, the hydrophilicity increased significantly, and the floatability of talc and pyrite was inhibited. This shows that guar gum has a strong inhibitory effect on talc and pyrite, which is consistent with the single mineral flotation results.

The influence of the intermolecular assembly of the flotation agent on the wettability of the mineral surface was investigated by the wet contact angle test. The results are shown in figure 8.

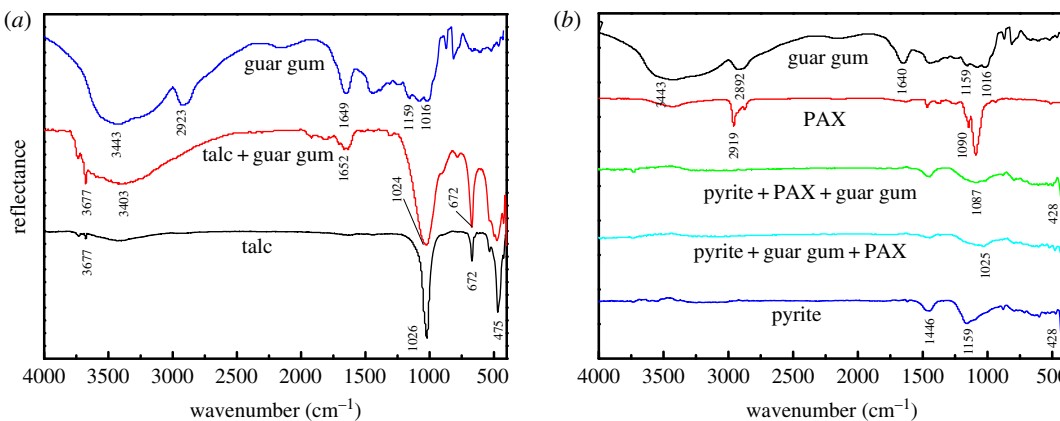

**Figure 9.** Infrared spectra of guar gum, PAX, talc and pyrite: (*a*) talc surface treated with and without guar gum; (*b*) pyrite surface treated with and without PAX and guar gum.

Figure 8*a* shows the change of the surface wettability of pyrite before and after the intermolecular assembly of the flotation agent. It can be seen that when no inhibitor is added, the surface contact angle of pyrite gradually increases with the increase in PAX dosage and the hydrophobicity gradually increases. When the amount of PAX is $2 \times 10^{-4} \, mol \, l^{-1}$, the contact angle of the pyrite surface is raised to 72°. In the system where the flotation agent and inhibitor coexist, guar gum significantly reduces the contact angle of pyrite. However, the contact angle of pyrite increases after the intermolecular assembly of the mineral surface flotation agent, and the hydrophobicity is restored.

Figure 8*b* shows the change of the wettability of the talc surface before and after the intermolecular assembly of the flotation agent. It can be seen that the contact angle of the talc is kept at a high level when no inhibitor is added, and the surface of the talc is hydrophobic. In the system in which the flotation agent and the inhibitor coexist, the guar gum significantly reduces the contact angle of the talc and increases the hydrophilicity of the surface of the talc. After the mineral surface flotation agent was assembled, the contact angle of the talc did not change significantly, and the hydrophilicity was maintained. Through the adjustment of the dosing agent and inhibitor dosing sequence, the hydrophobicity of the talc surface is reduced while the hydrophobicity of the pyrite surface is increased, thereby expanding the difference in the surface hydrophobicity between pyrite and talc.

## 3.4. FT-IR spectroscopy

Figure 9*a* shows the infrared spectrum of talc before and after the effect of guar gum. In the infrared spectrum of guar gum, $3443 \, cm^{-1}$ is the –OH stretching vibration, $2923 \, cm^{-1}$ is the –CH$_2$ stretching vibration and $1649 \, cm^{-1}$ is ring stretching vibration of the carbon oxygen six-membered ring. The characteristic peak at $1016–1159 \, cm^{-1}$ is the stretching vibration of the CO bond [26].

After the interaction of talc and guar gum, the talc showed an –OH stretching vibration peak at $3403 \, cm^{-1}$, which showed a larger deviation from the –OH stretching vibration peak in the guar gum, indicating that the guar gum passed the –OH [27,28]. The group is adsorbed on the surface of the talc. After the interaction with guar gum, the infrared spectrum of talc showed a new absorption peak at $1652 \, cm^{-1}$, which is the ring stretching vibration peak of the carbon–oxygen six-membered ring in the guar gum after adsorption [13].

Figure 9*b* shows the infrared spectrum of the adsorption of the agent on the pyrite surface before and after the intermolecular assembly of the flotation agent. Before the intermolecular assembly of the flotation agent, the order of addition of the agent is guar gum/PAX, and the adsorption peak of $1026 \, cm^{-1}$ appears in the infrared spectrum of the pyrite, corresponding to the CO bond stretching vibration of guar gum [29], and the guar gum is adsorbed on the PAX ion. After the intermolecular assembly of the flotation agent, the order of addition of the agent is PAX/guar gum, and the adsorption peak of $1087 \, cm^{-1}$ appears in the infrared spectrum of pyrite, corresponding to the C=S bond stretching vibration characteristic peak of PAX [27]. It can be seen that after the intermolecular assembly of the flotation agent, the adsorption of the collector PAX on the surface of the pyrite is enhanced, and the adsorption of the inhibitor guar gum on the surface of the pyrite is weakened.

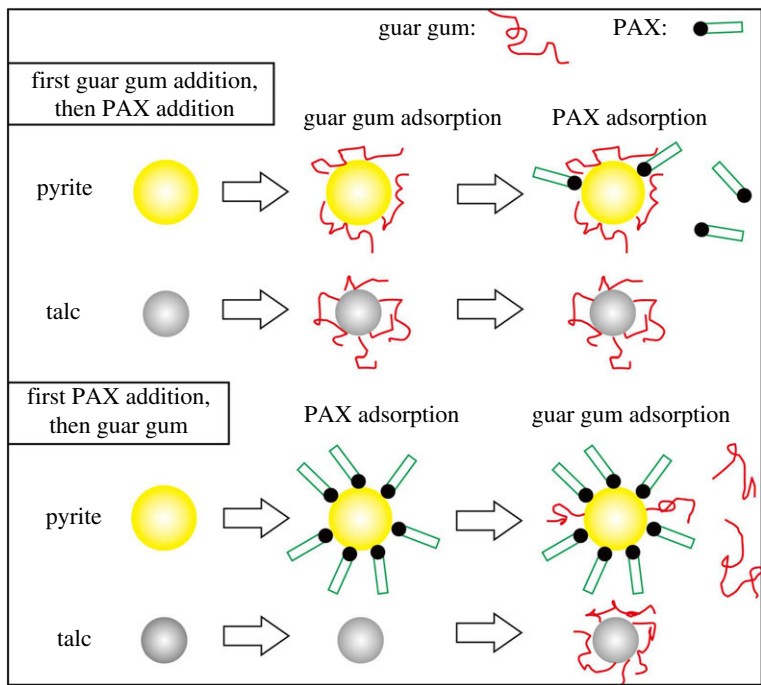

**Figure 10.** A competitive adsorption model of the collector and inhibitor on talc and pyrite surfaces.

## 3.5. Competitive adsorption model

In view of the difference in the adsorption capacity of collectors and inhibitors on the surface of two minerals, by adjusting the intermolecular assembly of the flotation agent at the mineral solid–liquid interface, the collector is preferentially adsorbed on the surface of the pyrite, increasing the surface of the pyrite. Hydrophobicity prevents pyrite from further adsorbing inhibitors containing hydrophilic groups, and achieves the selective inhibition of talc by inhibitors. This paper realizes the intermolecular assembly of the mineral surface flotation agent by adjusting the order of action of the agent; that is, adding the collector preferentially and then adding the inhibitor. Figure 10 shows the model of the interaction of flotation agents on the mineral surface.

# 4. Conclusion

We investigated the selective inhibition of talc in a sulfide ore flotation system by micro-flotation tests, adsorption measurements, contact angle measurements and infrared spectrum analysis and propose a competitive adsorption model between guar gum and PAX on the solid–liquid interface. The main conclusions are as follows:

(1) Organic high molecular polymers of guar gum are effective inhibitors of talc flotation, but also inhibit the flotation of sulfide ore.
(2) Guar gum adsorbs on the mineral surface through the active groups (–COOH and –OH) on the molecular chain to reduce the hydrophobicity of the mineral surface.
(3) The collector (PAX) is only adsorbed on the sulfide ore surface, while the inhibitor (guar gum) is adsorbed on both the sulfide ore and talc surface.
(4) The assembly of the reagent molecules at the mineral/water interface is achieved by changing the order in which the reagents are added. The collector is preferentially applied to the sulfide ore surface, thereby preventing further adsorption of the inhibitor on the sulfide ore surface and enhancing the selective adsorption capacity of the inhibitor on the mineral surface.

Data accessibility. The datasets supporting this article have been uploaded as part of the electronic supplementary material.
Authors' contributions. W.X. set up the degradation system. T.L. and W.X. carried out the micro-flotation tests and adsorption amount measurements. T.L. and W.Y. measured the contact angle and FT-IR analysis. W.X. and T.L. contributed to analysis data and wrote the draft of the manuscript. Finally, W.Y. revised the manuscript.

Competing interests. We declare that there are no competing interests.

Funding. This work was supported by the National Natural Science Foundation of China (grant nos. 51304150 and 51474169).

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
