## [Reviewer comments · Royal Society Open Science]

Review History

RSOS-191133.R0 (Original submission)

Review form: Reviewer 1

Is the manuscript scientifically sound in its present form?

Yes

Are the interpretations and conclusions justified by the results?

Yes

Is the language acceptable?

Yes

Do you have any ethical concerns with this paper?

Yes

Have you any concerns about statistical analyses in this paper?

No

Recommendation?

Accept with minor revision (please list in comments)

Comments to the Author(s)

The present job reports the impact of molecular assembly between guar gum and PAX for the flotation separation of pyrite and talc using micro-flotation tests. The adsorption mechanisms of guar gum and PAX on the pyrite and talc surfaces were explored through several analysis measurements. It was found that PAX preferentially adsorbs on the pyrite surface, thereby preventing guar gum from adsorbing on the pyrite surface and achieving selective inhibitor of talc by guar gum. These contents should represent the progresses of treated complex copper sulfide nickel ore and should be interested to the readers in the related fields. I recommend its acceptance for publication after minor revisions. Detailed comments are listed below.

1. The paper addresses a very good topic of molecular assembly between collectors and inhibitor on minerals. A main concern is that the reference papers related to molecular assembly are relatively fewer. The authors are suggested to consider quoting very recent findings of molecular assembly between collectors and inhibitor. For example, *Minerals Engineering*, 2018, 127: 42-47; *Powder Technology*, 2019, 345: 35-42.
2. In the "Introduction", the innovation and significance of the paper are not clearly explained, please rewrite.
3. Typing and language expression must be revised and improved in the whole paper.
4. Line 16, page 3, "2 g of the ore sample was placed...". The number should not appear in the first position.
5. Line 57, page 3, "Gol" should be "guar gum".
6. In this paper, there are many "In summary", and they need to make appropriate modifications.

Review form: Reviewer 2

Is the manuscript scientifically sound in its present form?

Yes

Are the interpretations and conclusions justified by the results?

Yes

Is the language acceptable?

No

Do you have any ethical concerns with this paper?

No

Have you any concerns about statistical analyses in this paper?

No

Recommendation?

Accept with minor revision (please list in comments)

Comments to the Author(s)

This work presents interesting results for the inhibitor of talc flotation, particularly the inhibitory effect of sulfide mineral and talc. It should be interested to the readers in the related fields. I recommend its acceptance for publication after minor revisions.

- 1, In Introduction it would be useful to discuss shortly some available data on the effect of guar gum on copper sulfide nickel ore (and/or other minerals) flotation.

- 2, What were the droplet sizes used for contact angle measurement? What are the errors (standard deviations) associated with each measurement? Did authors look into the effect of droplet size on the measured contact angles?
- 3, Moreover, the manuscript would benefit from being read by a native speaker.

Decision letter (RSOS-191133.R0)

13-Aug-2019

Dear Dr Xiao

On behalf of the Editors, I am pleased to inform you that your Manuscript RSOS-191133 entitled "The effect of molecular assembly between collectors and inhibitor on the flotation of pyrite and talc" has been accepted for publication in Royal Society Open Science subject to minor revision in accordance with the referee suggestions. Please find the referees' comments at the end of this email.

The reviewers and handling editors have recommended publication, but also suggest some minor revisions to your manuscript. Therefore, I invite you to respond to the comments and revise your manuscript.

- Ethics statement

- Data accessibility

If you wish to submit your supporting data or code to Dryad (<http://datadryad.org/>), or modify your current submission to dryad, please use the following link:
<http://datadryad.org/submit?journalID=RSOS&manu=RSOS-191133>

- Competing interests

- Authors' contributions

All submissions, other than those with a single author, must include an Authors' Contributions section which individually lists the specific contribution of each author. The list of Authors should meet all of the following criteria; 1) substantial contributions to conception and design, or

acquisition of data, or analysis and interpretation of data; 2) drafting the article or revising it critically for important intellectual content; and 3) final approval of the version to be published.

- Acknowledgements

- Funding statement

Because the schedule for publication is very tight, it is a condition of publication that you submit the revised version of your manuscript before 22-Aug-2019. Please note that the revision deadline will expire at 00.00am on this date. If you do not think you will be able to meet this date please let me know immediately.

- 1) A text file of the manuscript (tex, txt, rtf, docx or doc), references, tables (including captions) and figure captions. Do not upload a PDF as your "Main Document";

- 2) A separate electronic file of each figure (EPS or print-quality PDF preferred (either format should be produced directly from original creation package), or original software format);
- 3) Included a 100 word media summary of your paper when requested at submission. Please ensure you have entered correct contact details (email, institution and telephone) in your user account;
- 4) Included the raw data to support the claims made in your paper. You can either include your data as electronic supplementary material or upload to a repository and include the relevant doi within your manuscript. Make sure it is clear in your data accessibility statement how the data can be accessed;
- 5) All supplementary materials accompanying an accepted article will be treated as in their final form. Note that the Royal Society will neither edit nor typeset supplementary material and it will be hosted as provided. Please ensure that the supplementary material includes the paper details where possible (authors, article title, journal name).

on behalf of Prof R. Kerry Rowe (Subject Editor)
openscience@royalsociety.org

Associate Editor Comments to Author:

We have received two positive reviews on your manuscript, although both have raised a handful of concerns that will need to be addressed before your paper can be formally accepted. We would like to invite you to revise your paper as requested and then resubmit for further consideration.

When resubmitting your paper, please ensure that you have provided a point-by-point response, detailing how you have addressed each of the reviewers' individual concerns. In addition to this, we would recommend sending your paper onto a recognised language editing service, or passing your paper along to a native speaker to check the language used in your submission. For information about language editing services endorsed by the Royal Society, please see the following link: <https://royalsociety.org/journals/authors/language-polishing/>

Reviewer comments to Author:

Reviewer: 1

Comments to the Author(s)

The present job reports the impact of molecular assembly between guar gum and PAX for the flotation separation of pyrite and talc using micro-flotation tests. The adsorption mechanisms of guar gum and PAX on the pyrite and talc surfaces were explored through several analysis measurements. It was found that PAX preferentially adsorbs on the pyrite surface, thereby preventing guar gum from adsorbing on the pyrite surface and achieving selective inhibitor of talc by guar gum. These contents should represent the progresses of treated complex copper sulfide nickel ore and should be interested to the readers in the related fields. I recommend its acceptance for publication after minor revisions. Detailed comments are listed below.

1. The paper addresses a very good topic of molecular assembly between collectors and inhibitor on minerals. A main concern is that the reference papers related to molecular assembly are relatively fewer. The authors are suggested to consider quoting very recent findings of molecular assembly between collectors and inhibitor. For example, *Minerals Engineering*, 2018, 127: 42-47; *Powder Technology*, 2019, 345: 35-42.
2. In the "Introduction", the innovation and significance of the paper are not clearly explained, please rewrite.
3. Typing and language expression must be revised and improved in the whole paper.
4. Line 16, page 3, "2 g of the ore sample was placed...". The number should not appear in the first position.
5. Line 57, page 3, "Gol" should be "guar gum".
6. In this paper, there are many "In summary", and they need to make appropriate modifications.

Reviewer: 2

Comments to the Author(s)

This work presents interesting results for the inhibitor of talc flotation, particularly the inhibitory effect of sulfide mineral and talc. It should be interested to the readers in the related fields. I recommend its acceptance for publication after minor revisions.

- 1, In Introduction it would be useful to discuss shortly some available data on the effect of guar gum on copper sulfide nickel ore (and/or other minerals) flotation.
- 2, What were the droplet sizes used for contact angle measurement? What are the errors (standard deviations) associated with each measurement? Did authors look into the effect of droplet size on the measured contact angles?
- 3, Moreover, the manuscript would benefit from being read by a native speaker.

Author's Response to Decision Letter for (RSOS-191133.R0)

See Appendix A.

Decision letter (RSOS-191133.R1)

22-Aug-2019

Dear Dr Xiao,

I am pleased to inform you that your manuscript entitled "The effect of molecular assembly between collectors and inhibitors on the flotation of pyrite and talc" is now accepted for publication in Royal Society Open Science.

on behalf of Prof R. Kerry Rowe (Subject Editor)
openscience@royalsociety.org

Follow Royal Society Publishing on Twitter: [@RSocPublishing](https://twitter.com/RSocPublishing)

Appendix A

Dear Andrew Dunn and reviewers:

Thank you very much for your careful review and constructive suggestions with regard to our manuscript “**The effect of molecular assembly between collectors and inhibitor on the flotation of pyrite and talc**” (Manuscript ID: RSOS-191133). We have revised the manuscript strictly according to your kind advices and reviewer’s detailed suggestions. Those comments are helpful for us to revise and improve our paper. The main corrections in the paper and the responds to the reviewer’s comments are as following. All changes to the original manuscript were shown in yellow font. We appreciate for Editors/Reviewers’ warm work earnestly, and hope that the corrections will meet with approval. Please feel free to contact us with any questions and we are looking forward to your consideration. Thank you very much for all your help.

Best regards

Sincerely yours

Wei Xiao

Responds to the comments:

Reviewer: 1

Comments to the Author(s)

The present job reports the impact of molecular assembly between guar gum and PAX for the flotation separation of pyrite and talc using micro-flotation tests. The adsorption mechanisms of guar gum and PAX on the pyrite and talc surfaces were explored through several analysis measurements. It was found that PAX preferentially adsorbs on the pyrite surface, thereby preventing guar gum from adsorbing on the pyrite surface and achieving selective inhibitor of talc by guar gum. These contents should represent the progresses of treated complex copper sulfide nickel ore and should be interested to the readers in the related fields. I recommend its acceptance for publication after minor revisions. Detailed comments are listed below.

Reply: Thanks for your positive comments about our manuscript.

Comment 1: The paper addresses a very good topic of molecular assembly between collectors and inhibitor on minerals. A main concern is that the reference papers related to molecular assembly are relatively fewer. The authors are suggested to consider quoting very recent findings of molecular assembly between collectors and inhibitor. For example, *Minerals Engineering*, 2018, 127: 42-47; *Powder Technology*, 2019, 345: 35-42.

Reply: Thanks for your valuable suggestion. We have already cited some relevant literatures in Introduction section. The revised contents are as follows:

3. Fang S, Xu L, Wu H, Shu K, Xu Y, Zhang Z, Chi R, Sun W. 2019 Comparative studies of flotation and adsorption of Pb(II)/benzohydroxamic acid collector complexes on ilmenite and titanite. *Powder Technol.* 345, 35-42. (doi:10.1016/j.powtec.2018.12.089)
4. Wu, H.; Tian, J.; Xu, L.; Fang, S.; Zhang, Z.; Chi, R. 2018 Flotation and adsorption of a new mixed anionic/cationic collector in the spodumene-feldspar system. *Miner. Eng.* 127, 42-47. (doi:10.1016/j.mineng.2018.07.024)

Comment 2: In the “Introduction”, the innovation and significance of the paper are not clearly explained, please rewrite.

Reply: Thanks for your valuable suggestion. We have already rewritten the innovation and significance of this paper in the “Introduction” in the revised manuscript.

The innovation of this paper is to achieve the selective inhibition effect of organic polymer inhibitor (guar gum) on talc by using the difference of molecular assembly between collectors and inhibitors on sulfide minerals and gangue minerals. This study will help to understand the molecular assembly between the collector and the corrosion inhibitor to further process complex talc-type copper-sulfide-nickel ore.

Comment 3: Typing and language expression must be revised and improved in the whole paper.

Reply: We have revised the whole manuscript carefully in the revised version. In addition, we have asked the person who is skilled in English to check the English.

Comment 4: Line 16, page 3, “2 g of the ore sample was placed...”. The number should not appear in the first position.

Reply: Thank this referee’s correction. We have corrected the errors in the revised manuscript.

Comment 5: Line 57, page 3, “Gol” should be “guar gum”.

Reply: We have corrected this mistake in the revised manuscript.

Comment 6: In this paper, there are many “In summary”, and they need to make appropriate modifications.

Reply: Thanks for your valuable suggestion. We have made appropriate modifications to two “In summary” in the revised manuscript.

Reviewer: 2

Comments to the Author(s)

This work presents interesting results for the inhibitor of talc flotation, particularly the

inhibitory effect of sulfide mineral and talc. It should be interesting to the readers in the related fields. I recommend its acceptance for publication after minor revisions.

Reply: Thanks for your positive comments about our manuscript.

Comment 1: In Introduction it would be useful to discuss shortly some available data on the effect of guar gum on copper sulfide nickel ore (and/or other minerals) flotation.

Reply: Thanks for your valuable suggestion. We have added some available data to discuss the effect of guar gum on copper sulfide nickel ore flotation in the revised manuscript. The revised contents are as follows:

Bicak et al.⁸ compared the inhibitory effects of guar gum and CMC on pyrite in sulphide flotation, and found that guar gum had a strong inhibitory effect on pyrite at low dosage. Therefore, under conventional flotation conditions, guar gum is difficult to be used as an inhibitor of gangue minerals with better natural flotation, such as sulphide ore and talc.

Comment 2: What were the droplet sizes used for contact angle measurement? What are the errors (standard deviations) associated with each measurement? Did authors look into the effect of droplet size on the measured contact angles?

Reply: The droplet used for contact angle measurement contained 50 μL of milli-Q water, and we controlled the droplet size by a calibrated syringe. The equilibrium contact angle of the water droplet on mineral surface was measured in four different positions, and the mean value was reported as the result.

But the four measurement values were all shown in Fig. 7, and the measured error between the mean value and measurement value was $\pm 3^\circ$. We thought the error was expressed more directly than the standard deviations, due to record the all measurement value.

The effect of droplet size on the measured contact angles was reported in many literatures, and we also found that the droplet size was an important factor for the contact angle measurement. So, the droplet size was strictly controlled to the same in all measurements.

According to your questions, we have revised the contact angle measurement section in the revised manuscript.

Comment 3: Moreover, the manuscript would benefit from being read by a native speaker.

Reply: We have revised the whole manuscript carefully in the revised version. In addition, we have asked the person who is skilled in English to check the English.

We have modified the manuscript carefully according to the comments. All the changes have been marked in red font. Thank you again for your valuable comments and suggestions. The manuscript has been resubmitted to your journal. We are looking forward to your positive response. Please contact us anytime if needed, the E-mail is xiaowei2015@csu.edu.cn (Wei Xiao).

Best Regards.

Wei Xiao